

# Soil and Biomass Carbon Storage is Much Higher in Central American than Andean Montane Forests

Cecilia M. Prada[1], Katherine D. Heineman[1,2], Maria J. Pardo[3], Camille Piponiot[4], James W. Dalling[1,3]

[1]Department of Plant Biology, University of Illinois, Urbana, IL 61801, USA
[2]San Diego Zoo Wildlife Alliance, Escondido, CA, 92025, USA
[3]Smithsonian Tropical Research Institute, Panama, Panama
[4]UR Forests and Societies, Cirad, Univ Montpellier, Montpellier, France

*Correspondence to*: Cecilia M. Prada (ceprada@gmail.com)

**Abstract.** Tropical montane forests (TMF) play a key role in the global carbon (C) cycle and in climate regulation by sequestering large amounts of above and belowground carbon. Elevation gradients in TMF have helped reveal the influence of environmental factors on C stocks. However, the influence of elevation and soil nutrient availability on C stocks has not been evaluated for mixed arbuscular and ectomycorrhizal (EM) associated forests in the Neotropics. We estimated aboveground biomass (AGB), coarse wood debris (CWD), and soil C based on field inventories in ten 1-ha plots along an elevational gradient from 880 to 2920 m a.s.l varying in relative abundance of EM-trees in western Panama. Trees ≥ 10 cm diameter at breast height (DBH) and CWD ≥ 10 cm diameter were measured to calculate biomass and necromass. Soil C to 1 m depth was estimated. Furthermore, climate and edaphic characteristics were described for each plot to evaluate the influence on these variables on each C pool. AGB, downed CWD and soil C were strongly positively correlated with elevation. We found exceptionally high AGB, up to 574.3 Mg ha$^{-1}$, and soil C, up to 577.9 Mg ha$^{-1}$ at higher elevations. Variation in total CWD within and among plots was high ranging from 14.75 to 326.5 Mg ha$^{-1}$. After controlling for elevation, neither nutrient availability nor EM-dominance had an effect on AGB or soil C. Nonetheless, high AGB at high elevations was attributed to the presence of *Quercus* species. Remarkably high soil C at high elevations might be a consequence of reduced decomposition rates associated with lower temperature, or geological history, where repeated volcanic eruptions buried surface soil organic layers. Our results highlight large regional uncertainty in C pool estimates in Neotropical montane forests, with potentially large underestimates for Central American C stocks.

## 1 Introduction

Tropical forests store ~40-55% of the total carbon (C) in terrestrial ecosystems (Pan et al., 2011; Meister et al., 2012; Pan et al., 2013), and therefore play a key role in the global C cycle and climate regulation (Malhi and Grace, 2000; Keenan and Williams, 2018; Mitchard, 2018). This carbon is sequestered in several pools, consisting of live biomass, necromass (i.e. coarse woody debris), and soil. Estimating of the size of each pool is critical to evaluating uncertainty in future global C budget predictions.

While it is well known that climate and edaphic processes influence C stocks (Moser et al., 2011; Clark et al., 2002; Girardin et al., 2010), mycorrhizal interactions have also been shown to have a significant impact on C cycling at the global scale (Phillips et al., 2013; Averill et al., 2014; Steidinger et al., 2019). Evaluating the effects of these factors on tropical forest C stocks is an important step in predicting future atmospheric CO$_2$ levels (Keenan and Williams, 2018).

Studies along elevational gradients in the tropics, especially in the Andes, have been shown to be powerful natural experiments to understand how the environment directly and indirectly influences processes related to C cycling (Girardin et al., 2014; Malhi et al., 2017; Girardin et al., 2010; Nottingham et al., 2015a). Factors such as temperature, cloud cover, precipitation,





nutrient availability, and species composition change as elevation increases, affecting aboveground biomass (AGB), woody debris and the soil C stock. For AGB, a meta-analysis found at lower elevations AGB can be higher (Muller-Landau et al. 2020). However, low values of AGB have been found in the Amazon below 500 m a.s.l (Quesada et al., 2012). High temperatures in the

lowlands are typically associated with fast plant growth rates; in the Amazon, studies have shown faster growth rate is associated with lower mean wood density (Malhi et al., 2004; Baker et al., 2004), suggesting low AGB. However, it is important to note that low wood density is not always associated with a reduction in AGB (Phillips et al., 2019b). On the other hand, for soil C stocks, in upper montane forests, C can accumulate at low temperatures by decreasing microbial decomposition (Nottingham et al., 2015a).

While elevational gradients can help us understand how temperature influences ecosystem carbon stocks, additional potentially confounding factors vary with elevation. For instance, cloud cover varies from 57% at low elevations up to 89% at high elevations (Muller-Landau et al., 2020) and precipitation can peak at middle or high elevations (Malhi et al., 2010). In montane forests soil nutrient availability can vary over short geographic scales, associated with variation in parent material and substrate age (Prada et al., 2017). Soil nutrient availability affects forest productivity in the tropics, but its effect on AGB is less

clear, showing a positive, negative, or no relationship (Muller-Landau et al., 2020). Furthermore, soil nutrient availability influences tree tissue nutrient content and therefore litter quality and potentially wood decomposition rate (Aerts and Chapin, 1999). Recalcitrant tissues with low nutrient concentrations decompose slowly and increase soil carbon sequestration due to their long residence time in soil (De Deyn et al., 2008). However, there is also evidence that fast-decomposing labile tissues can remain in the soil as mineral-associated soil organic matter, and thus increase soil C (Cotrufo et al., 2013).

While environmental factors play a critical role in forest structure and function, plant mutualisms with microbes also strongly impact soil C storage. Ectomycorrhizas (EM) and arbuscular mycorrhizas (AM) are thought to influence C stocks via effects on decomposition (De Deyn et al., 2008; Averill et al., 2014; Phillips et al., 2013). Studies have found that forest trees associated with EM fungi promote slow C cycling compared to trees associated with AM fungi (Soudzilovskaia et al., 2019; Phillips et al., 2013). In temperate forests this may be partly due to lower quality litter in EM trees than in AM trees, resulting in

slower decomposition (Craig et al., 2018; Keller and Phillips, 2019). Additionally, the stratification of soil C storage with soil depth also appears to depend on mycorrhizal type; soil under EM trees stores more C in the surface layers while soil under AM trees stores more carbon in deep layers (Craig et al., 2018).

In this study we estimated the size of three C pools: AGB, coarse wood debris (CWD), and soil C in ten 1-ha plots along an elevational gradient in western Panama and evaluated the influence of climate, edaphic and biotic factors on each pool.

To our knowledge this is the first study in EM-associated forests that integrates both the effect of soil nutrient availability and elevation on C stocks in the Neotropics. Studies in western Panama provide new insights into the drivers of C storage because unlike many Andean forests, the dominant ecological group shifts with elevation, from AM-dominated forests at low elevations to EM-dominated forests at high elevation. We hypothesized that i) a reduction in temperature with increasing elevation results in reduced growth rates, increased wood density and therefore reduced forest turnover rate and CWD, and potentially also AGB

(if stand height also declines with elevation as in other neotropical forests (Girardin et al., 2010; Unger et al., 2012); ii) decreasing nitrogen availability with elevation (Marrs et al., 1988) would further favor trees with conservative nutrient cycling strategies (i.e., high wood density and low nutrient concentration; Soong et al., 2020), also resulting in lower canopy turnover rates and CWD. However, wood from low soil nutrient availability sites has lower nutrient concentrations (Heineman et al., 2016), resulting in slower decomposition, which may allow CWD and soil C to accumulate; iii) forests associated with EM-tree

species have high overall C storage, EM-tree taxa that form monodominant stands in tropical forests have previously been associated with low nutrient soils (Hall et al., 2020), tend to have conservative nutrient cycling strategies (Phillips et al., 2013)



and have been associated with soils with higher C:N (Lin et al., 2017; Averill et al., 2014). Therefore, following our previous hypothesis, we expected the presence of EM-trees to increase AGB relative to those with AM-tree species at similar elevation. Additionally, if EM associations either reduce decomposition rates due to differences in litter quality or nutrient immobilization
(McGuire et al., 2010; Read and Perez-Moreno, 2003), then we predicted higher accumulation of soil C in EM-dominated forests. Finally, our ground sampling across forests with contrasting edaphic conditions and species composition allowed us to compare C estimates for our sites with previously published data from other neotropical forests along elevation gradients, and to compare AGB estimates for our plots with estimates derived from a combination of airborne Light Detection and Ranging (LiDAR) and more limited ground truthing datasets that did not account for structural and compositional variation between
lowland and montane forests (Asner et al., 2013).

## 2 Materials and methods

### 2.1 Study Site

The study was conducted in western Panama along an elevational gradient ranging from 880 to 2920 m a.s.l within the Fortuna Forest Reserve (Fortuna), the southern edge of the adjacent Palo Seco Forest Reserve (8˚45'N, 82˚13'W) and at Finca El Velo
(8˚49"11'N, 82˚28"58'W), on the northeastern slope of Volcan Barú National Park (Barú). The elevational gradient extends from super-humid premontane forests (700-1000 m a.s.l) and lower montane forest (1000-1500 m a.s.l) (Holdridge 1947) in Fortuna, to mid-elevation montane forests (1700-2000 m a.s.l) and upper montane cloud forests (>2000 m a.s.l) at Barú. At Fortuna, mean annual rainfall ranges between 4600 and 6300 mm (Prada et al., 2017; Dalling et al., 2021), while at 2300 m a.s.l at Barú, mean annual temperature is 14 ˚C and mean annual rainfall from 2019 to 2021 was 2260 mm. For this study, ten
permanent 1-ha plots were established: six plots at Fortuna in 2003 and four at Barú in 2015, where all trees with ≥ 5 cm diameter at breast height (DBH) were measured and identified to the lowest taxonomic level possible. When a species identification was not determined a morphospecies was designated. In this study to facilitate comparisons with other published datasets only trees ≥ 10 cm DBH were included in the analyses.

Across the elevational gradient, plots vary in relative abundances of different taxonomic and functional groups that are known to
have important ecological roles in tropical forests, including trees associated with EM and AM, N-fixers, and palms (Table 1). Based on literature, trees were classified depending on mycorrhizal type (Corrales et al., 2018; Soudzilovskaia et al., 2020; Averill et al., 2019) and N-fixing ability (Huss-Daniel, 1997; Sprent, 2009). The representation of ecological groups (EM, N-fixers and palms) was calculated as their percentage contribution to the total basal area of each plot.

**Table 1.** Summary of characteristics of ten 1-ha plots along an elevational gradient in western Panama including trees >10 cm DBH. Percent of basal area of species that form ectomycorrhizal associations (EM), N-fixing, and palm taxa. FFR= Fortuna Forest Reserve, VBNP = Volcan Baru National Park.

| Plot | Plot code | Site | Type of forest | Elevation (m a.s.l) | No. trees | No. stems* | Basal area (m²ha⁻¹) | % EM | % N-fix | % Palms |
|---|---|---|---|---|---|---|---|---|---|---|
| PaloSeco‡ | PS | FFR | Mixed | 880 | 630 | 646 | 31.2 | 2.4 | 17.4 | 2.3 |
| AltoFrio§ | AF | FFR | Mixed | 1100 | 1019 | 1042 | 44.6 | 7.1 | 8.5 | 0.0 |
| ChorroA† | CA | FFR | Palm-dominated[b] | 1100 | 979 | 1002 | 32.6 | 9.2 | 0.4 | 41.7 |
| HondaA† | HA | FFR | Mixed/ EM-dominated[c] | 1155 | 800 | 804 | 41.6 | 24.0 | 4.4 | 0.4 |
| Samudio* | SAM | FFR | Mixed | 1215 | 1049 | 1071 | 33.7 | 0.78 | 10.4 | 0.8 |
| Hornito¤ | HOR | FFR | Mixed | 1330 | 681 | 694 | 55.2 | 2.2 | 2.2 | 0.0 |



| Mirador [þ] | MIR | VBNP | Mixed | 1987 | 457 | 477 | 46.7 | 1.1 | 4.9 | 0.0 |
| CasitaA [þ] | CASA | VBNP | EM-dominated[a] | 2248 | 646 | 686 | 50.5 | 74.1 | 0.4 | 0.0 |
| Quetzal [þ] | QUE | VBNP | EM-dominated[a] | 2599 | 543 | 564 | 59.0 | 48.1 | 3.2 | 0.04 |
| Copete [þ] | COP | VBNP | EM-dominated[a] | 2923 | 630 | 728 | 63.0 | 63.3 | 7.3 | 0.0 |

Geology: [‡] mafic volcanic; [§] undifferentiated volcanics; [†]rhyolite; [*] mafic/rhyolite; [¤] dacite; [þ] Holocene lahar flow. Forest type: [a] *Quercus*-dominated; [b] *Colpothrinax aphanopetala*-dominated*; [c] *Oreomunnea mexicana*-dominated *No. of stems used for the analyses includes multiple stems of individual trees

## 2.2 Soil sampling variables

Soil data are described in detail for the plot network in Fortuna (Dalling et al., 2021; Turner and Dalling, 2021). For this study, soil data were used for six plots from Fortuna plots network collected in July 2008 or July 2010 (Prada et al., 2017) and from four Barú plots, collected in May 2016. Soil from each plot was analyzed for bulk density (BD), pH, resin extractable P (ResinP), extractable inorganic N ($NH_4$ and $NO_3$), extractable cations (Al, Ca, Fe, K, Mg, Zn), effective cation exchange capacity (ECEC), total exchangeable bases (TEB), and total N and P. Following the protocols described in Prada et al. (2017), soil samples were collected from the surface 0–10 cm of soil after removing the litter layer. Soil samples were collected in a regular grid (from every other 20 × 20 m subplot) within each 1-ha plot for a total of thirteen sampling locations.

## 2.3 Climate variables

To evaluate the climate regime across the elevational gradient we obtained data from the CHELSA database (http://chelsa-climate.org) at 30 arc-sec spatial resolution (Karger et al., 2017) for each plot. We obtained ten variables that have been shown to be important in determining forest structure: MAT (mean annual temperature), MATvar (mean temperature range), MATmax (max temperature), MATmin (min temperature), TS (temperature seasonality), MAP (mean annual precipitation), MPdry (mean of driest month), MPwet (mean of wettest month), ISO (isothermality) and MDR (mean diurnal range). Additionally, we extracted the average climatic water deficit (WD) data from a global climate layer for the long-term at 2.5 arc-minute resolutions (http://chave.ups-tlse.fr/pantropical_allometry.htm). Climatic WD describes seasonal moisture limitation, with very negative WD values indicating strongly seasonally water-stressed sites.

## 2.4 Carbon stock calculations

### 2.4.1 Aboveground biomass and aboveground carbon

We used the BIOMASS R package to estimate AGB (Réjou-Méchain et al., 2017); functions in this section are all part of this package. For each plot, all stems ≥ 10 cm DBH (including multi-stemmed trees) were measured and identified to genus, species or morphospecies. The DBH used for the analyses was the value from the last re-census measured from Fortuna and Barú (2018 and 2015 respectively). We estimated the height of a subsample of 78 ± 11 trees from each plot using a triangulation approach, measuring the angle to the tree base, the angle to treetop and the distance to the tree. The angles were measured using a manual clinometer (PM5/360PC, Suunto, Finland). To estimate the height of each tree in each plot, we created a plot height-diameter model using the *log 1* equation in the function *modelHD()* (Fig. S1). *Log 1* was the best model for seven of the plots based on the AIC index compared to the other models proposed in Réjou-Méchain et al (2017). To estimate the wood density of each tree we measured the density of 117 species across all plots, with an average of eight individuals measured for each species (Fig. S1). Wood density in the field was calculated using the water displacement method (Chave, 2006). This dataset was incorporated into



the function *getWoodDensity*(), assigning a density to each tree based on our field wood density, then the nearest species, genus or family measurement if the field data were missing. Trees without any identification (23% of the total trees) were assigned the average plot level measurement. Finally, to estimate the AGB we used Chave et al. (2014) allometric Eq. (1):

$$AGB = 0.0673 * (\rho * H * D^2)^{0.976} \tag{1}$$

Where $\rho$ is the wood density, H is the height, and D is the DBH of each tree. We used the function *AGBmonteCarlo*() to estimate

95 % confidence intervals at a plot level.

To compare the distribution of AGB values between lowland and montane forests across the region we used 212 plots from twelve studies in the Neotropics, including ten plots from this study (Table S1). The distributions of AGB values among forest types and elevation classes were assessed by bootstrapping the AGB with 1000 resamples. Finally, we compared AGC values obtained in this study, using the same *AGBmonteCarlo*() function, but for the carbon value (Réjou-Méchain et al., 2017)

that use a C ratio of 47.1 (Thomas and Martin, 2012) with those estimated from airborne Light Detection and Ranging (LiDAR) data extracted for the same locations and at the same 1-ha spatial resolution (Asner et al., 2013, 2021).

### 2.4.2 Downed coarse wood debris

To estimate the downed CWD (DCWD) we used a line-intersect sample method following Gora et al. (2019) and Larjavaara & Muller-Landau (2011). Downed CWD was defined as wood debris fallen to ground level that was ≥ 10 cm diameter and crossed

the transect. For each plot, we measured DCWD along twenty-two transects of 0.1 km. For the PS plot (Table 1) we only conducted 10 transects given the difficulty of traversing this plot. Transects were oriented south to north and east to west every 10 m within and including the edge of the plot. For each piece of DCWD we used a large caliper to measure the diameter at the point the transect crossed the sample. Using a qualitative classification method, each sample was classified into three decomposition categories 1) 'Hard', if the sample was hard to the touch, 2) 'Medium', if the sample was still somewhat solid but

soft to the touch, 3) 'Soft', if the sample was visually rotten and collapsed easily. For each decomposition category and for each plot we took destructive samples consisting of a cross-section that was weighed, measured, and dried to estimate the necromass (Gora et al., 2019). With the destructive samples we used linear regressions to estimate the necromass of the samples, where the necromass was a function of diameter, decomposition category, and plot. All cross-section masses were corrected by the random angle correction factor ($\pi$/2), summed and divided by the transect length. For the final estimates of the DCWD we calculated the

95% confidence intervals by bootstrapping over unit samples (10 m; Gora et al., 2019).

### 2.4.3 Standing coarse wood debris

To estimate the standing CWD (SCWD) we used a plot approach following Gora et al. (2019) and Larjavaara & Muller-Landau (2011). Standing CWD was defined as dead trunks with a DBH ≥ 10 cm. We measured SCWD in five 20×20 m subplots, for a total of 0.2 ha area sampled in each plot. For each SCWD sample we measured the DBH, height (H), classified it in the same

decomposition categories as the DCWD, and additionally classified it in two qualitative branch categories: 1) With branches, if the sample retained between 20-100% of branches, and 2) without branches, if the sample retained < 20% of branches.

We estimated the necromass of the standing dead wood in two ways. For standing dead wood with branches we used the BIOMASS package (Réjou-Méchain *et al*., 2017) using Eq. (1) in this study. In this case $\rho$ was calculated using the DCWD destructive samples. Since the wood densities differed between DCWD decomposition categories ($F_{(2,326)} = 87.35$, $P = <0.001$),

wood density was calculated as the average of each decomposition category by plot. Height was the total height of the sample measured in the field. Necromass was calculated as 87.5% of the total original AGB estimated (Gora et al., 2019) using Eq. (1).





For standing dead wood without branches we calculated the volume and density of each sample. We used a taper function to estimate the diameter at the top using the Kozak *et al*. (1969) equation in Cushman et al. (2014) and the volume was calculated as a truncated cone for samples ≥ 3 m high. For samples < 3 m high we calculated the volume as a cylinder. Densities were

calculated in the same way as branched SCWD. We summed the necromass of all the samples in all categories and we divided by the total area to estimate the SCWD by plot. We calculated the 95% confidence intervals by bootstrapping over unit samples (20×20 m subplots; Gora et al., 2019).

**2.4.4 Soil Carbon**

To estimate the soil C stocks in each plot we collected soil in four depth increments from 0 to 100 cm at a subset of five of the

thirteen locations sampled for soil nutrients. For the Fortuna network plots the 50-100 cm depth soil C value was based on a single estimate from profile pits adjacent to the plots excavated to 1.5–2.0 m (Turner & Dalling, 2021). The 0–10 cm, 10–20 cm and 20–50 cm depth samples were taken with a 5 cm diameter core, and for 50–100 cm depth samples we used a 6.25 cm diameter corer. Total C stock in each depth was calculated as the product of the total C%, bulk density, and the increment of each depth. Total C% was determined by elemental analysis (Thermo Flash 1112 analyzer, Bremen, DE) and bulk density following

Prada et al. (2017). The 95% confidence interval of the total C stock was calculated by bootstrapping over locations for each plot.

**2.5 Data analysis**

All environmental variables were standardized to a mean of zero and standard deviation of one to perform the analyses.

**2.5.1 Plot characterization**

To explore and visualize the variation of environmental variables among plots, soil and climate variables were analyzed separately with a principal component analysis (PCA).

**2.5.2 Effect of soil and mycorrhizal type on trees' wood density**

One-way ANOVA was used to evaluate differences among plots soil characteristics and trees wood density. We included mycorrhizal type as a fixed effect and species as random effect. *P*-values of the main effects were estimated using model

simplification (Crawley, 2013).

**2.5.3 Effect of environmental variables and mycorrhizal type on C stocks**

To examine the relationship between environmental variables and mycorrhizal type across the different carbon pools we used a structural equation model (SEM) to partition the variance of responses. An *a priori* model was hypothesized based on our predictions and simple linear regressions between environmental variables and each carbon pool (Fig. S2, Table S2). A second

set of PCA was performed excluding resin-extractable P, $NH_4$ and temperature variables since we were interested in those direct effects; the first PCA scores were then used as environmental and edaphic parameters in the SEM (Fig. 5 a, b). To correct for collinearity the model included the correlation between variables that are highly correlated (r > 0.5, Table S3). Representation of EM-associated trees was incorporated in the model as the percent of plot basal area represented by EM-tree species (Table 1). We then used SEM to evaluate the indirect and direct effect of each independent variable on each carbon pool, using the

piecewiseSEM R package (Lefcheck, 2016). To discriminate between models, we followed the approach of Glassmire et al.,



(2020) for adding and dropping variables. Akaike's information criterion (AIC) was used to compare the alternative models. The best model was the most parsimonious; the fit of the best model was evaluated using Fisher's C statistic (Lefcheck, 2016). Lower values of Fisher's C value indicate smaller differences between the model and the data. Standing CWD was not included in the SEM analysis since it was not correlated with any of the abiotic or biotic factors or to the other carbon pools.

## 3 Results

### 3.1 Edaphic and climate variation

Edaphic characteristics were highly heterogenous across plots – $NH_4$ varied six-fold, $NO_3$ twelve-fold, and resin-extractable P varied 87-fold (Table S4). The first two axes of the first soil PCA (Fig. 1a) explained 68.9% of the variance in the edaphic variables among plots. In the first axis, two plots, MIR and AF were grouped together by variables associated with high fertility (i.e., cation exchange capacity (ECEC)). The Fortuna plots CA, HA, PS and SAM grouped together and were associated low pH and high Fe in the soil. The second axis described soil variation with elevation (Fig. 1a). The plots at the highest elevation are at the bottom of the figure and are associated with high values of resin P.

The first two axes of the climate PCA (Fig. 1b) explained 87.9% of the variance in the climate variables among plots. The first axis grouped plots by the sites Fortuna and Barú and shows the association between climate variation and elevation (Fig. 1b). The plots at the highest elevations are at the right of the figure and are associated with low mean annual temperatures (MAT) and high mean precipitation during the driest month (MPdry). The second axis groups together three plots, CA, HA, and PS, which are associated with high values of mean diurnal range (MDR). The plots HOR and AF group together related to high mean precipitation during the wettest month (MPwet). Although there is high variation in seasonal water stress (WD) (Table S4), its contribution to climate variation was low (Fig. 1b).

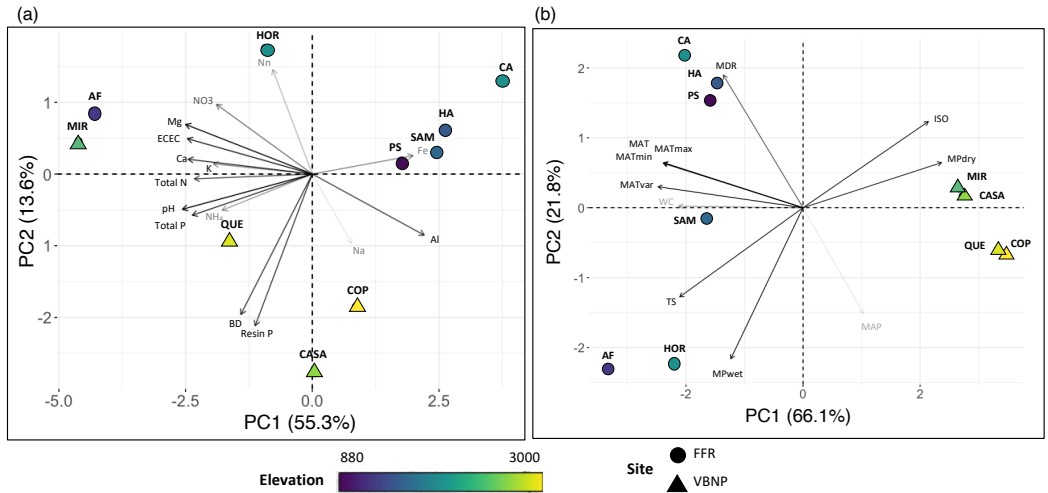

**Figure 1.** Principal component analysis (PCA) for (a) soil variables 0-10 cm depth, and (b) climate variables of ten 1-ha plots (Table 1) in an elevational gradient in Panama. Direction and length of vectors indicates the effect of the variable. Color represents the elevation of the plot, and the symbols represent the site. BD = bulk density, ResinP = resin extractable P, ECEC = effective cation exchange capacity, TEB = total exchangeable bases, MAT = mean annual temperature), MATvar =mean temperature range, MATmax = max temperature, MATmin = min temperature, TS = temperature seasonality, MAP = mean





annual precipitation, MPdry = mean of driest month, MPwet = mean of wettest month, ISO = isothermality, MDR = mean diurnal range, WD= water deficit, FFR= Fortuna Forest Reserve, VBNP=Volcan Baru National Park. Plot name codes are given in Table 1.

**3.2 Total carbon stock estimates**

Total carbon stocks for the ten plots ranged from 249.3 to 882.1 Mg ha$^{-1}$ (Table 2, Fig. 2a). Aboveground carbon and soil C were the most important carbon pools, contributing 23–50% and 32.5–70.7%, respectively, of the total carbon budget of the forest (Fig. 2b). The contribution of CWD to the total carbon budget was highly variable, ranging from 3.8–44% between plots. For example, for QUE and HOR the plot total CWD was nearly 10 times smaller than AGC, while for HA the total CWD

contribution was higher than AGC (Fig. 2b).

**Table 2.** Total estimated carbon pools in Mg ha$^{-1}$ with 95% confidence intervals for ten 1ha plots in an elevational gradient for trees and CWD samples with diameter ≥10 cm. AGB = Aboveground biomass; AGC = Aboveground carbon, CWD = Coarse wood debris

| Plot | Plot code | Elevation | AGB | AGC | CWD | | | Soil C | Total Carbon |
| | | | | | Downed | Standing | Total CWD | | |
|---|---|---|---|---|---|---|---|---|---|
| PaloSeco | PS | 880 | 231.7 (211.6, 255.7) | 109.1 (99.7, 119.3) | 12.14 (8.72,15.59) | 9.03 (1.68,19.09) | 21.17 | 119.0 (76.2, 170.8) | 249.3 |
| AltoFrio | AF | 1100 | 311.2 (292.4, 336.4) | 146.5 (136.1, 157.1) | 10.35 (7.14,14.1) | 86.06 (17.56,178.98) | 96.41 | 260.5 (215.5, 307.2) | 503.4 |
| ChorroA | CA | 1100 | 186.2 (174.6, 200.5) | 87.5 (81.6, 93.9) | 7.35 (5.79,9.1) | 52.72 (15.28,119.86) | 60.07 | 233.4 (143.6, 296.7) | 381.0 |
| HondaA | HA | 1155 | 370 (334.1, 420) | 173.6 (156.4, 194.9) | 13.49 (10.48,17.08) | 313.02 (20.69,757.09) | 326.51 | 241.3 (138.7, 318.5) | 741.4 |
| Samudio | SAM | 1215 | 264.8 (249.6, 281.3) | 124.9 (117.7, 132.8) | 16.1 (13.39,18.98) | 39.64 (2.27,103.53) | 55.74 | 193.3 (115.3, 274.9) | 374.0 |
| Hornito | HOR | 1330 | 408.3 (375.3, 445.6) | 192.7 (176.8, 212.2) | 11.38 (7.77,15.37) | 3.37 (0.92,5.83) | 14.75 | 174.3 (133.6, 213.8) | 381.7 |
| Mirador | MIR | 1987 | 387.6 (346.9, 425.2) | 182.6 (165.1, 202.4) | 20.37 (15.7,25.94) | 13.83 (0.66,31.13) | 34.2 | 524.7 (428.7, 640.6) | 741.5 |
| CasitaA | CASA | 2248 | 468.3 (432.4, 512.8) | 220.5 (203.2, 239.8) | 21.32 (16.94,26.23) | 50.4 (3.78,108.02) | 71.72 | 317.3 (273.4, 361.2) | 609.6 |
| Quetzal | QUE | 2599 | 574.3 (524.8, 628.6) | 271.4 (247.6, 299.1) | 26.97 (21.06,33.87) | 7.9 (2.97,14.49) | 34.87 | 473.1 (400.2, 544) | 779.4 |
| Copete | COP | 2923 | 518.3 (482.8, 559.1) | 244.2 (227.0, 262.9) | 48.9 (35.81,63.01) | 11.08 (1.83,26.02) | 59.98 | 577.9 (474.2, 642.9) | 882.1 |




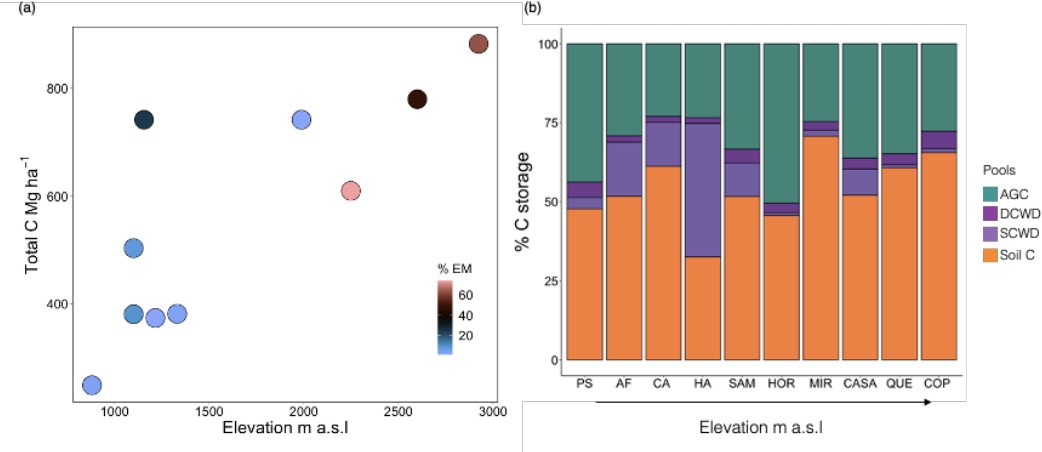

**Figure 2.** Total C stocks (a) and contribution (b) of the different pools to the total carbon stocks for ten 1-ha plots in an elevational gradient in western Panama. AGC = Aboveground carbon; DCWD= Downed coarse wood debris; SCWD = Standing coarse wood debris; % EM= percent of plot basal area occupied by EM trees.


### 3.2.1 *Aboveground Biomass*

To estimate the AGB, 7434 trees (from 91 plant families) and 780 tree heights were measured across all of the ten plots. Aboveground biomass varied three-fold among plots, ranging from 186.2 to 574.3 Mg ha$^{-1}$ (Table 2). Different taxa contributed to AGB in different plots, with important contributors including Fagaceae (57.0-83.8%), Sapotaceae (22.1-38.8%) and Arecaceae

(37.2%). In addition to taxon differences, contributions of ecological groups to AGB also differed across plots. EM-trees ranged from 0.98-83.8% of AGB across plots. EM-trees in the genus *Quercus* were the most important contributors to AGB at the highest elevation plots (57-83.8%), and at the HA plot, where EM-trees including *Oreomunnea mexicana* (Juglandaceae) accounted for 24% of basal area and 18.4% of the AGB. N-fixing trees accounted for 0.25-19.2% of the total AGB of the plots. Palms were the most important ecological group in the CA plot, accounting for 37.2 of AGB, whereas palms contributed < 1% of

total AGB in the remaining plots. EM-trees had higher wood density compared to AM-trees ($F_{(1,259)}$= 15.32, d.f. = 1, $p < 0.001$) where mean wood density (g cm$^3$) for EM-trees was 0.71 ± 0.02 and for AM-trees 0.54 ± 0.008 (SE).

Combining data from this study and eleven published studies in Neotropical forests (Table S1) we found that variance in AGB for montane forests (1037-3537 m a.s.l) was exceptionally high, compared to lowland forests (41-1000 m a.s.l) (Fig. S3a, b). We also found that aboveground carbon density at our field sites was substantially under-estimated in a previous

approach using LiDAR overflight data (Asner 2013, 2021). LiDAR based values for AGC were reliant on conversion factors generated from ground-truthing lowland forests and were between two and six times lower than our field-based estimates (Fig. 3).





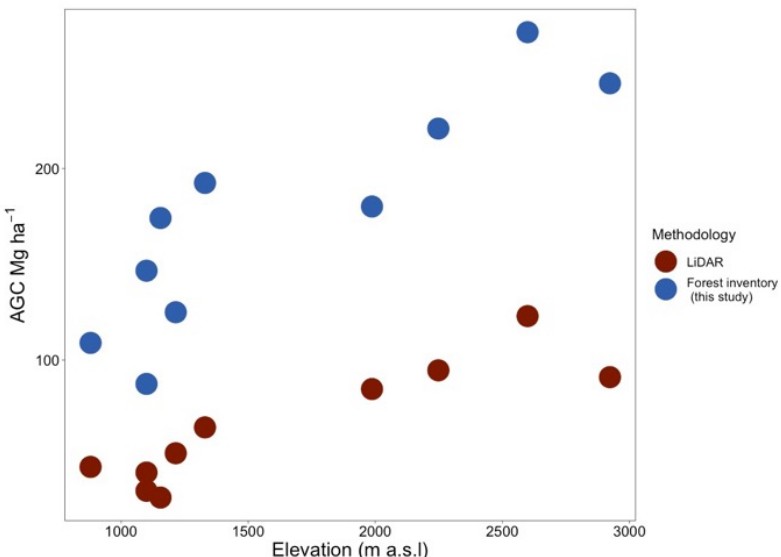

**Figure 3.** Variation in aboveground carbon (AGC) among ten 1-ha plots in western Panama using forest inventories (blue) and
(red) airborne light detection and ranging (LiDAR) data.

### 3.2.2 *Coarse wood debris*

In total, 2037 samples of DCWD were found in 20.8 km of transect, and 232 samples of SCWD were measured in 0.2 ha per
plot. Total CWD varied across plots, ranging from 14.75 to 326.51 Mg ha$^{-1}$ (Table 2). Across plots there was no clear pattern in
the contribution of DCWD and SCWD to the total CWD. For PS, HOR, MIR, QUE and COP dead wood was mainly stored in
the form of DCWD, contributing 2.7-5.5% to the total carbon of the plot, while for AF, CA, SAM, CASA and HA, SCWD was
the main CWD pool, contributing 8.2 to 42.2%.

### 3.2.3 *Soil carbon*

Total soil C calculated from 0-100 cm depth differed among plots, and was similar in magnitude and variance to AGB, ranging
from 119.0 to 577.9 Mg ha$^{-1}$ (Table 2). There were differences among plots ($F_{(9,181)}$ = 10.9, $p$ = < 0.001) and depth categories
($F_{(3,181)}$= 31.5, $p$ = <0.001). Plots at the highest elevation (1900 – 2923 m a.s.l) accumulated notably large amounts of carbon in
the deeper soil layers (50-100 cm), ranging from 92.8 to 246.3 Mg ha$^{-1}$ (Fig. S4). The percent of C in each depth relative to the
total C% in each plot was not correlated with % EM at any depth (Fig. S5). However, at 10-20 cm depth, percent of C was
positively correlated with the percent of basal area contributed by EM-trees (% EM; $r^2$ = 0.36, $p$ < 0.05,), however this effect was
not significant after controlling for elevation.

### 290 3.2.4 *Effect of elevation on abiotic and biotic factors, and carbon pools*

Most climate variables were strongly correlated with elevation (Table S5); variables related to temperature (MAT, MATmax,
MATmin and MATvar), water deficit and mean diurnal range were negatively correlated with elevation. Precipitation during the
driest month was positively correlated with elevation. Among edaphic variables only resin P was correlated with elevation,
increasing with elevation (r = 0.88, $p$ < 0.001). Additionally, % EM was also highly positively correlated with elevation (r =





0.80, $p < 0.001$). Resin P and % EM were therefore also highly correlated with temperature (r = -0.89, $p < 0.001$ and r = -0.76, $p$ < 0.05, respectively).

Aboveground biomass and soil C showed a significant positive relationship with elevation ($r^2 = 0.73$ $p < 0.001$; $r^2 =$ 0.76, $p < 0.001$, respectively, Fig. 4a,b). Total CWD was not correlated with elevation ($r^2 = -0.06$, $p =$ NS), even after removing plot HA, an outlier, ($r^2 = 0.001$, $p =$ NS). However, DCWD was positively correlated with elevation ($r^2 = 0.76$, $p < 0.001$, Fig.

4c). Low values of AGB, DCWD and soil C are significantly correlated with high values of variables related with temperature, while soil C decreased with increasing wood density ($r^2 = 0.76$, $p < 0.001$), however, this effect was not significant after controlling for elevation.

Among the AGB components, elevation was correlated with basal area but not with wood density or canopy height (Fig. 4d-f). Basal area was positively correlated with elevation, more than doubling over the elevation range of the study ($r^2 = 0.65$, $p <$

0.001). The number of stems per plot tended to decrease with elevation, but was not significant ($r^2 = 0.18$, $p =$ NS).



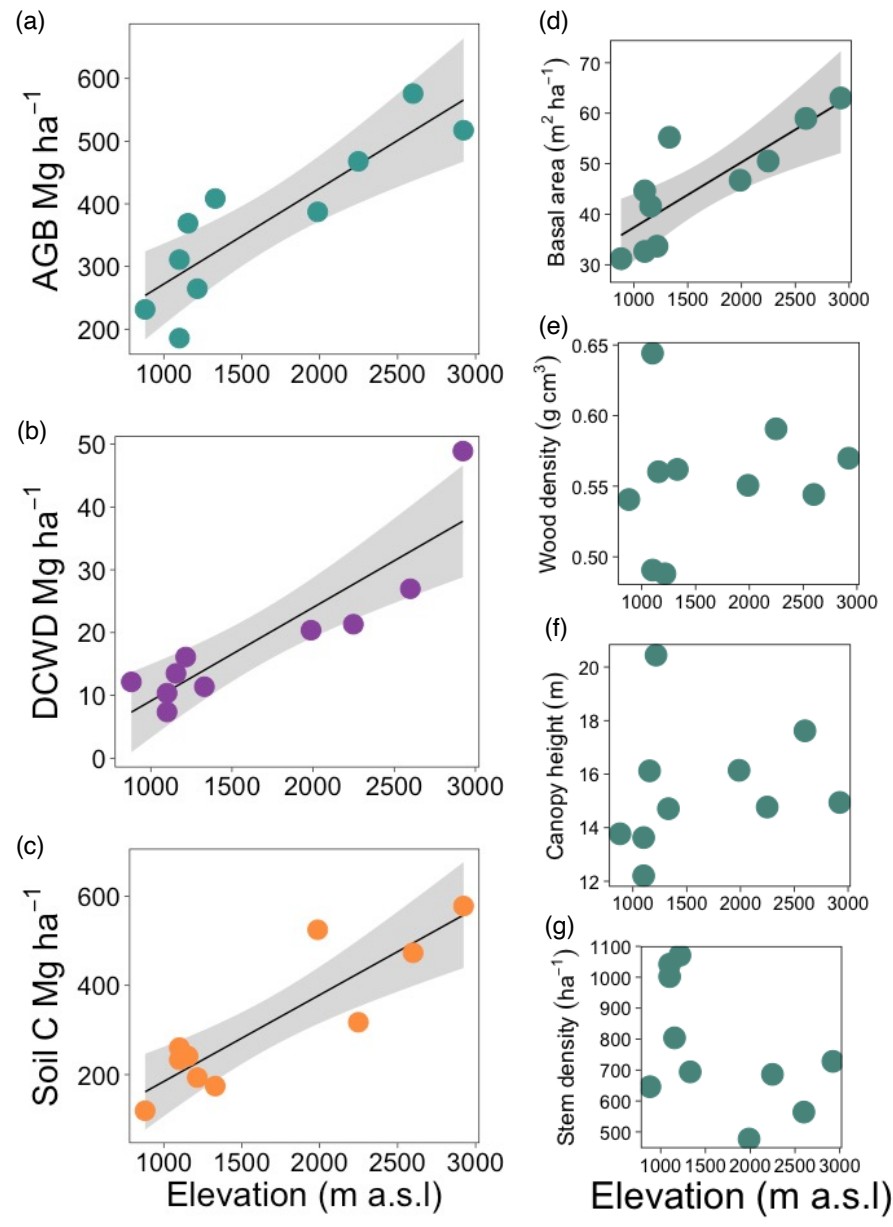

**Figure 4.** Effect of elevation on carbon pools (a – c) and aboveground biomass components (d – g) for ten 1ha plots in an elevational gradient in western Panama. Lines represent significant relationships and shaded areas represent the 95% confidence intervals. Each point is the mean value for each plot except for (d) that is the total basal area of the plot. AGB = Aboveground biomass; DCWD= Downed coarse wood debris.





315        From the structural equation model (SEM) that included soil, climate variables, AGB, DCWD, and soil C we found temperature was the variable that best predicted AGB and soil C, after controlling for soil and other climate parameters (Fig. 5). Temperature was significantly negatively correlated with AGB and soil C. Resin P, $NH_4$ and other soil parameters (represented in the first PCA axis, Fig. 6 a), and % EM showed a significant correlation with DCWD. Downed CWD decreased significantly with high values of resin P, $NH_4$ and soil parameters and increased with high values of % EM. Additionally, % EM was

significantly positively correlated with resin P. The effect of climate parameters other than temperature (represented in the first PCA axis, Fig. 6b), did not have a significant effect on C pools.

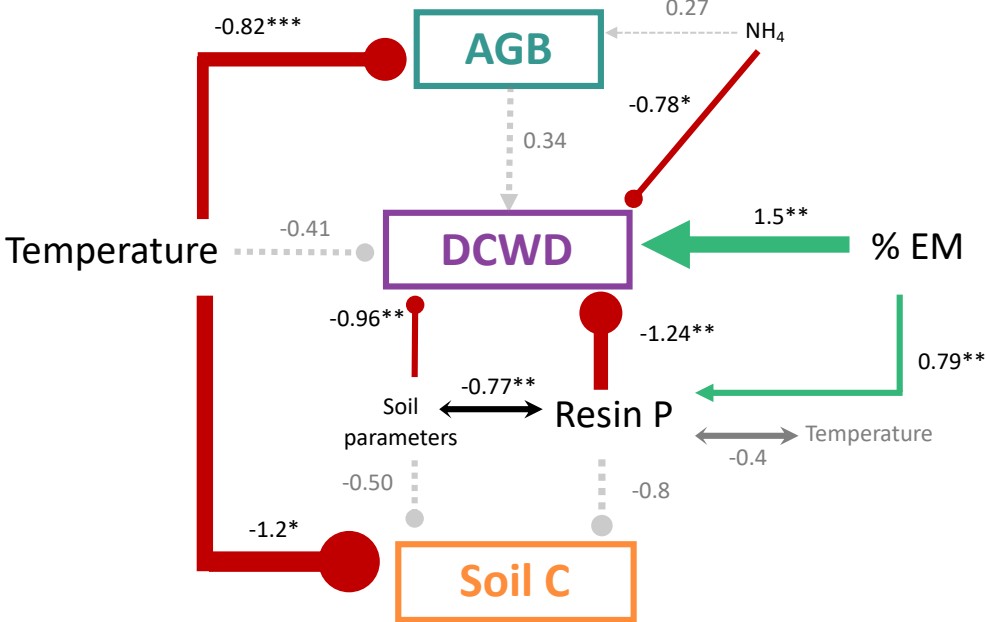

$C_{(12)}$= 11.41; p=0.494; AIC= 55.4

**Figure 5.** Structural equation model to evaluate the effect of environmental and edaphic variables on the different carbon pools in a montane forest. Soil and edaphic parameters are represented by the first and second axes of a PCA for each group of variables where resin P and $NH_4$ were not included. The best model was selected based on Akaike's information criterion (AIC) comparing alternative models. Positive effects are in green and represented by arrows and negative effects in red with blunt-ended lines; arrow width represents the relative importance of each factor. Gray dashed arrows represent nonsignificant

relationships. Lines with double arrows represent correlations between variables. Numbers are the standardized path coefficients. AGB = Aboveground biomass; DCWD= Downed coarse wood debris; % EM= percent of plot basal area occupied by EM trees.



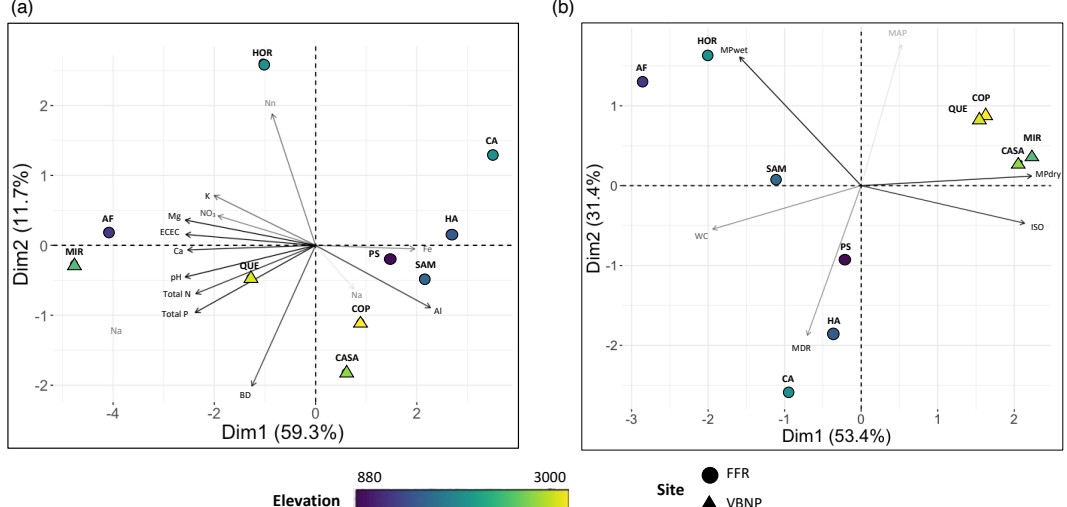

**Figure 6.** Principal component analysis (PCA) used for structural equation model (SEM) for (a). soil parameters 0-10 cm depth excluding resin P and NH$_4$ and (b). climate parameters excluding variables related to temperature. Direction and length of vectors indicates the effect of the variable. Color represents the elevation of the plot and the symbols represent the site. For abbreviations see Fig. 1 caption.

## 4 Discussion

Exceptionally high carbon storage was observed along this elevational gradient in Panama. In contrast to most other published data from elevational gradients in TMFs, we found total carbon stocks were strongly *positively* correlated with elevation – a pattern associated with the dominance of the EM-tree genus *Quercus* at most of the high elevation plots. Temperature was a strong predictor of AGB and soil C. In contrast, we found resin P and % EM explained more variation in downed CWD (DCWD) stocks than temperature; DCWD accumulation was associated with increasing % EM and decreasing resin P and NH$_4$. Estimates of standing CWD (SCWD) were poorly constrained in this study, with exceptionally high values found in one site. We found high soil C was also associated with high elevation sites. We predicted soil C storage would be associated with EM-trees, however we found no effect of % EM on total soil C. The plot with patches dominated by the EM-tree *Oreomunnea mexicana* had intermediate values of soil C while high elevation plots at the Barú site both with and without EM-associated trees showed the highest values of soil C, possibly reflecting both reduced rates of decomposition, and volcanic activity that buried C rich surface soils beneath layers of volcanic ash in the past.

### 4.1 Carbon stocks of Panamanian montane forests in a regional and pantropical context

Values for AGB are highly variable across tropical forests. In Neotropical lowland forests, plot inventories of trees ≥ 10 cm DBH indicate that AGB ranges from 160.5 Mg ha$^{-1}$ at La Selva, Costa Rica (Clark & Clark, 2000), 114 to 172 in the Área de Conservación Osa, Costa Rica (Hofhansl et al., 2020), 262 on Barro Colorado Island, Panama (Chave et al., 2003), and up to 397 ± 30 in the central Amazon of Brazil (Nascimento and Laurance, 2002). Previous studies along elevational gradients have shown no consistent pattern between elevation and AGB, either in the Neotropics (Fig. 7a), or in meta-analyses of Neotropical, African and SE Asian forests (Cuni-Sanchez et al. 2021). In contrast, our study showed a significant increase in AGB with elevation, reaching a maximum value just a few hundred meters beneath the tree line on Volcan Barú. Comparing this study with four other Neotropical elevational gradients indicates sites in Colombia (González-Caro et al., 2020) and Panama (this study) share





relatively high AGB at mid-elevations (1500-2000 m) compared to Peru (Malhi et al., 2017), Ecuador (Moser et al., 2008) and Venezuela (Vilanova et al., 2018) (Fig. 7a). However, at high elevations AGB in our study was between three and seven times higher than AGB reported from these other gradients. Our Panama sites, however, do share high AGB with African montane forest plots spanning a similar elevation range. A recent meta-analysis of AGB for 226 African montane forest plots spanning a similar elevation range to our study found on average 326.75 Mg ha$^{-1}$ (± 28.5 Mg ha$^{-1}$ 95% CI; Cuni-Sanchez et al. 2021), a

value that is similar to our Panama study sites (372.07 ± 39.7 Mg ha$^{-1}$).

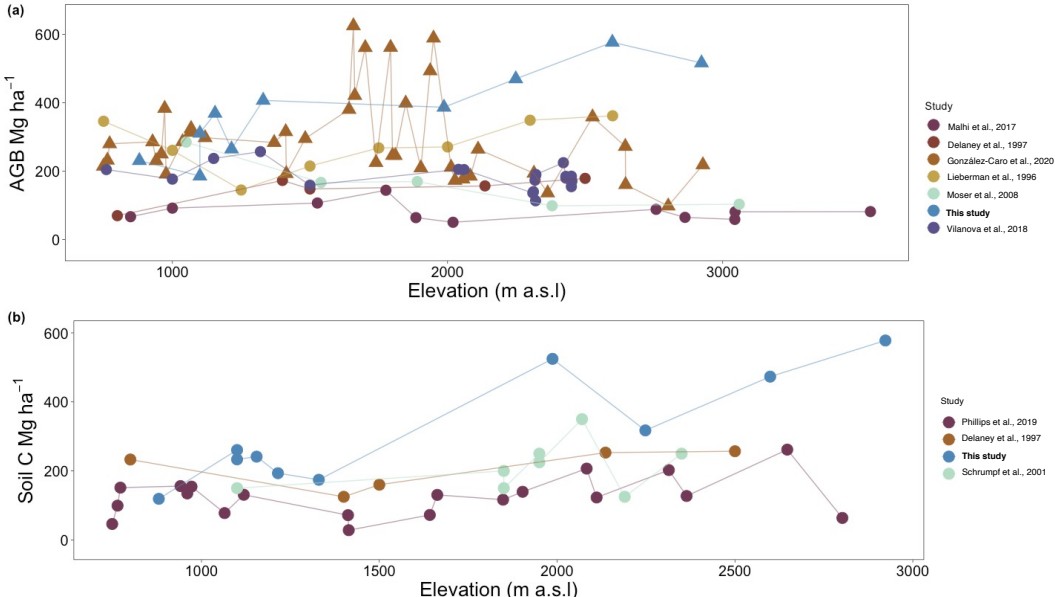

**Figure 7.** Comparison of above-ground biomass (a) and soil C (b) stocks among studies in the Neotropics. (a) Points represent AGB Mg ha$^{-1}$ in one plot for five studies; symbols indicate sites with presence of EM dominated forests (triangle) or absence

(circles). (b) Points represent Soil C Mg ha$^{-1}$ at 1 m depth for four studies. Studies were performed in Peru (Malhi et al., 2017), Ecuador (Moser et al., 2008; Schrumpf et al., 2001), Venezuela (Delaney et al., 1997; Vilanova et al, 2018), Costa Rica (Lieberman et al., 1996) and Colombia (González-Caro et al., 2020; Phillips et al., 2019a).

High levels of uncertainty in the magnitude and drivers of AGB across forests (Muller-Landau et al., 2020; Sierra et al.,

2007) presents considerable challenges for estimating AGC at large spatial scales. LiDAR provides a powerful tool that can be used to help estimate C stocks at high resolution across the globe (Réjou-Méchain et al., 2015; Asner et al., 2013). However, applying this approach requires adequate ground referencing to validate calculations. Asner et al. (2013) estimated and mapped aboveground carbon for the country of Panama using LiDAR. Although the map was validated with field data from different ecoregions of the country, no montane forest data were used. We found that the Asner et al. (2013) map consistently and

substantially underestimated the AGC present in our plots (Fig. 3), with implications for the national carbon inventory, given the large amount of montane forest cover in the country. These results highlight the need to include inventory data that incorporates elevation dependent changes in soils, climate and species composition that underpin elevational changes in AGB-canopy height relationships.



As in previous studies, soil C stocks were highly variable in this study. However, unlike other Neotropical studies (Fig. 7b) we found the highest soil C stocks in the highest elevation plots, supporting our initial hypothesis of increases in soil C with elevation. At our highest elevation site, COP, soil C to 1 m depth (578 Mg ha$^{-1}$) was more than four times higher than comparable measurements in lowland forests in Panama (133 Mg ha$^{-1}$, Cusack et al., 2018). Similarly, in our study, the soil C stock was five times higher in lower montane forest (1500-2000 m a.s.l) and two times higher in upper montane forest (2000-3600 m a.s.l), compared to studies in Colombia (Phillips et al., 2019a), Ecuador (Schrumpf et al., 2001) and Venezuela (Delaney

et al., 1997).

       Differences in methodologies and sample sizes across studies make comparisons of coarse wood debris difficult. Furthermore, variation in total CWD within and among plots in this study was exceptionally high. One plot, HA, had unusually high SCWD biomass (313 Mg ha$^{-1}$) with correspondingly large confidence intervals. However, the presence of a few large standing dead trees can greatly skew estimates of SCWD (Gora et al. 2019), necessitating large sample sizes to capture the true

mean pool size. Nonetheless, our study highlights the potential importance of this pool to the ecosystem C budget, and adequately constraining SCWD measurements to detect climate change effects on ecosystem carbon storage via elevated tree mortality rates.

**4.2 Responses of carbon pools to environmental and biotic factors**

**4.2.1 Aboveground biomass**

After controlling for elevation, we found no effect of soil fertility (measured as NH$_4$-N and resin P) on AGB. More generally, responses of AGB to soil nutrient availability are highly inconsistent among studies (Muller-Landau et al., 2020). For example, in lowland tropical forests in the Guiana Shield (Soong et al., 2020) AGB did not respond to soil P, whereas more broadly across the Amazon basin low AGB was found to be associated with high soil P (Quesada et al., 2012). Evaluating the effects of nutrient availability on AGB along elevational gradients is challenging as direct fertility effects may be masked by the strong correlations

between elevation, and therefore temperature, and soil N and P. Decreasing temperatures could have multiple effects altering AGB. Decreasing temperature can reduce growth rates and is associated with increased wood density (Muller-Landau, 2004; Chave et al., 2009), which may increase AGB. Alternatively lower temperatures are also associated with reduced stand height in other tropical forests, reducing AGB (Wilcke et al., 2008) . Here we found there is a positive relationship between elevation an AGB, despite wood density and canopy heigh not being correlated with elevation; showing basal area was the variable driving

AGB pattern. A positive relationship between elevation and basal area has been seen in other tropical forests (Unger et al., 2012; Lieberman et al., 1996; Clark et al., 2015), and in some cases a peak is attributed to the presence of *Quercus* spp (Muñoz Mazón et al., 2020), a genus associated with high wood densities (Cavender-Bares, 2019).

       While EM tree species had higher wood densities compared to AM trees, and made a large contribution to AGB, the observed positive relationship between elevation and AGB was not dependent on wood density (i.e., the pattern remained if all

species were given the same wood density value). In tropical forests in Colombia and Indonesia, relatively high AGB in highland forests is associated with a significant contribution of families such as Fagaceae and Podocarpaceae (Peña et al., 2018; Culmsee et al., 2010; González-Caro et al., 2020). Recent studies have shown that the evolutionary history of tree communities can contribute to tropical forest structure, with phylogenetic signal detected in AGB (de Aguiar-Campos et al., 2021; González-Caro et al., 2020). In Colombia, AGB at intermediate elevation (1800-2200 m a.s.l), is associated with a high proportion of temperate-

affiliated species (González-Caro et al. 2020). Among the most important of these is the genus *Quercus* (González-Caro et al., 2020; Peña et al., 2018). This aligns with our results, where *Quercus* was the largest contributor to AGB in three of the four highest elevation plots. *Quercus*, and related EM genera in the Fagaceae appear to be well adapted to the cool conditions of



tropical high elevation forests in SE Asia, Central America, and the northern Colombia, where individuals reach large sizes and high wood densities (Corrales et al. 2018, Cavender-Bares, 2019). However, while we found in this study that there was a strong
positive correlation between elevation and the percent of basal area contributed by EM trees, the effect of % EM on AGB was no longer significant after accounting for elevation. Notably, at the Barú site, the MIR plot had an AGB value comparable to *Quercus*-dominated plots despite very low % EM basal area (1% versus 48-74% in other Barú plots; Table 1). We propose that non-EM-trees (e.g. *Magnolia*, *Cornus*, *Cedrela*, *Cinnamomum*) may need to have similar architectural and wood density traits to be able to coexist with oak species. These results, therefore, contrast with AGB patterns at a smaller spatial scale in a 50 ha plot
in a temperate forest in China, where AGB was positively associated with the proportion of EM-basal area in the stand (Mao et al., 2019).

     The report of high AGC for 226 African montane forest plots (Cuni-Sanchez et al. 2021) is also achieved in the absence of EM-tree species (Fagaceae and Juglandaceae are absent from African mountains). The authors concluded that high montane forest carbon storage in Africa (70% higher than previous estimates for Neotropical forests) could reflect unique structural
characteristics of African montane forests resulting from the presence of large herbivores (elephants), or a low frequency of large disturbances (cyclones and landslides). A third potential explanation, where high AGB is associated with the presence of conifers (Podocarpaceae) was not supported by the data (Cuni-Sanchez et al. 2021). However, the same meta-analysis showed that SE Asian montane forests appear to have similar AGC to African forests. In conclusion, our results now suggest that rather than African forests having anomalously high carbon storage, central Andean forests may instead have anomalously low carbon
storage.

### 4.2.2 Coarse woody debris

Coarse woody debris in tropical forests is a critical component of the C cycle, with highly variable distributions in space and time, rendering it hard to study (Iwashita et al., 2013; Gora et al., 2019). The amount of CWD in the forest can depend on wood quality, climate and stand turnover rates (Chambers et al., 2001, 2000). Here we found that the responses of CWD to
environmental and biotic factors differed between CWD pools — downed CWD and standing CWD. Total CWD and SCWD were not correlated with elevation. After controlling for elevation, we found resin P, and $NH_4$ were negatively correlated with DCWD, and % EM positively correlated with DCWD. Previous studies in the Fortuna site have shown that wood from trees associated with low nutrient soils had higher wood density and lower wood nutrient concentrations than those of higher nutrient soils (Heineman et al., 2016). Although climate can be the best predictor of decomposition in Neotropical forests (Cusack et al.,
2009), litter quality is also important (Zhang et al., 2008). Low litter N and P concentration is associated with slow decomposition (Zhang et al., 2008), which may account for the high accumulation of DCWD on the forest floor on this study. Additionally, in the Amazon, high wood density is associated with lower decomposition rates (Chambers et al., 2000). In this study, DCWD accumulation with increasing % EM may be explained by higher values of wood density in EM-trees species compared to AM-trees species, and potentially also lower soil N availability in EM-dominated forests (Lin et al., 2017; Phillips
et al., 2013; Prada et al., 2022)

### 4.2.3 Soil Carbon

We found that temperature was the most important factor influencing soil C stocks. Soil C accumulated with increases in elevation, a pattern that has been reported previously for tropical montane forests (Kitayama and Aiba, 2002; de la Cruz-Amo et al., 2020; Schrumpf et al., 2011; Girardin et al., 2010; Moser et al., 2011). Continually cool temperatures at high elevation
decrease microbial activity, which results in a decrease in decomposition rates and mineralization of soil organic matter



(Davidson and Janssens, 2006; Nottingham et al., 2015b). Following the Microbial Efficiency-Matrix Stabilization (MEMS) framework (Cotrufo et al., 2013), slow decomposition rates at high elevations suggest C in the soil would be present as particulate soil organic matter (POM), consistent with the large organic horizon observed at the Barú sites. However, in this study we also found elevation was highly positively correlated with resin P. This strong correlation makes it hard to unravel the

true effect of both temperature and P on soil C stocks.

Additionally, in montane forests POM would be expected to increase in EM-forests (Craig et al, 2018). In this study, we cannot separate the effect of elevation and mycorrhizal association on POM accumulation. Increasing evidence also shows significantly higher soil C stock near the soil surface beneath EM-trees in both temperate and tropical forests (Averill et al., 2014; Craig et al., 2018; Lin et al., 2017; Steidinger et al., 2019), while greater soil C stock may occur at greater soil depths in

AM-forest (Craig et al., 2018). Mechanisms explaining differences in soil C stock have been attributed to differences in the enzymatic capacity of AM and EM species to mineralize organic matter, differences in litter quality, or in mycorrhizal carbon inputs (McGuire et al., 2010; Read and Perez-Moreno, 2003; Averill, 2016; Huang et al., 2022). In this study the effect of % EM on soil C may have been confounded by the correlation between % EM and elevation. Furthermore, recent work at a wider range of sites at Fortuna has shown that contributions of EM to soil organic matter varies widely among watersheds and is dependent

on soil pH (Seyfried et al., 2021). A clear example of how high soil C stocks are not necessarily dependent on the presence of EM-trees is provided by the MIR plot— a mixed forest in which EM-trees accounted for only 1.1% of basal area and yet had one of the highest soil C stocks. Nonetheless, given the proximity of *Quercus* trees to the MIR plot it is possible that soil C stocks at this site accumulated under past conditions when the site may also have been occupied by EM trees. An additional complication that affects numerous high elevation sites through Panama and Costa Rica is the impact of the eruptive history of a volcanically-

active mountain chain. Deep volcanic ash deposits, dating to 7 million years are apparent at the HA, SAM and especially the CA plots at Fortuna (Wegner et al., 2011; Turner & Dalling, 2021), while soil pits at Barú show soil C layers that are overlain by much more recent deposits of volcanic ash. At least four eruptions of Volcan Barú have occurred in the last 1600 years (Sherrod et al., 2008). The last episode, 500 years ago, may have included widespread tephra fallout, pyroclastic flows, and lahars across the area occupied by our plots (Sherrod et al., 2008). Volcanic eruptions result in the burial of the existing organic layer at the

time of the event, and can result in the long-term persistence of significant quantities of soil organic carbon (Chaopricha and Marín-Spiotta, 2014). For example on Mt. Kilimanjaro, Tanzania, soil organic carbon below 1 m depth ranged from 117 to 627 Mg ha[-1] in elevations from 2100 to 2800 m a.s.l (Zech, 2006). Particulate soil organic matter is expected to be more vulnerable to disturbance than C that is as mineral-associated organic matter MAOM (Poeplau et al., 2018) in lowland forests. Volcanic eruptions may protect POM, increasing soil C accumulation in these montane forests.

*4.2.3 Role of tropical elevational gradients in global C stocks*

Our results from the Fortuna - Barú gradient highlight the significance of these forests in the global C cycle, as they sequester large amounts of C in tree biomass and in the soil. While tropical forests represent a large proportion of the terrestrial carbon sink (Spracklen & Righelato, 2014; Duque et al., 2021), the importance of the contribution of montane forests (Cuni-Sanchez et al., 2021) and of the temperate family Fagaceae has only been fully recognized more recently (Culmsee et al., 2010; González-

Caro et al., 2020; Peña et al., 2018). Nonetheless, factors explaining the dynamics and accumulation of C in the soil are still ambiguous. Large uncertainty in C pool estimates suggests that further research is needed to correctly estimate and evaluate C stocks in tropical montane forests.



**Author contributions.** CMP and JWD conceived the ideas; CMP, KH and MP carried out field data measurements; CP

contributed with model analyses; CMP and JWD performed the general data analysis and wrote the paper.

**Competing interests.** The authors declare that they have no conflict of interest.

**Acknowledgements.** We want to thank the Smithsonian Tropical Research Institute for logistical support. Additional funding
was provided by the University of Illinois Urbana- Champaign Department of Plant Biology and the Patricia Peterson
Foundation. Autoridad Nacional del Ambiente (ANAM) provided research permits to undertake the study. We also want to thank

Luke Zehr, Lucas Hernandez, Evidelio Garcia, Fredy Miranda, Marlon Olmiranda and Marco Arturo Prada for field assistance;
Price Peterson and Carlos Espinosa for logistical support; and to the STRI Soil Lab for chemical analysis of soil samples.

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
