# Peer review of "Soil and Biomass Carbon Storage is Much Higher in Central American than Andean Montane Forests"

_EGUsphere, 2024_

## Author Response (AR1)

**REFEREE 1 COMMENTS**

In this study, Prada et al., conduct a very thorough field sampling in Panama to evaluate C stocks accounting for different AG fractions and wood debris, understand its drivers, and compare them among other tropical forests, studies, and methodologies. I believe the methodology is sound and it has potential for a valuable scientific contribution. Nonetheless, I believe that the interpretation behind the findings and some parts of the writing need to be strongly strengthened or reformulated.

General comments:

After reading the manuscript and incorporating the data provided, seems obvious to me that the main reason for the C stock gradient is the P availability/limitation. Many previous studies suggest that tropical forests, especially in America and in lowland forests, are largely P-limited. When transitioning to higher elevations, this P limitation gradually changes into N limitation. You can check Cunha et al., 2022, Vallicrosa et al., 2023, Wright 2019, among many others. The findings displayed in this paper seem to perfectly align with that previously reported gradient as stated in line 293 "Among edaphic variables only resin P was correlated with elevation, increasing with elevation ($r = 0.88$, $p < 0.001$))". Nonetheless, the authors' interpretation was the opposite, line 400 "After controlling for elevation, we found no effect of soil fertility (measured as NH4-N and resin P) on AGB". Of course, if you remove the variable elevation, which explains 88% of your variability, you end up with nothing to be explained. Here you are not performing a fertilization experiment so you cannot directly report limitations. Still, I think that the story you are telling needs to be reframed to match with that evidence.

Cunha, H.F.V., Andersen, K.M., Lugli, L.F. et al. Direct evidence for phosphorus limitation on Amazon forest productivity. Nature 608, 558–562 (2022). https://doi.org/10.1038/s41586-022-05085-2

Vallicrosa, Helena, Laynara F. Lugli, Lucia Fuchslueger, Jordi Sardans, Irene Ramirez-Rojas, Erik Verbruggen, Oriol Grau, et al. 2023. " Phosphorus Scarcity Contributes to Nitrogen Limitation in Lowland Tropical Rainforests." Ecology 104(6): e4049. https://doi.org/10.1002/ecy.4049

Wright, S. J. 2019. Plant responses to nutrient addition experiments conducted in tropical forests. Ecological Monographs 89(4):e01382. 10.1002/ecm.1382

**Response: Thank you for this comment. We agree with the reviewer that carbon stocks do not respond directly to elevation, but to a combination of mostly abiotic variables that correlate strongly with elevation. Identifying causative pathways is difficult given the strong covariation of these variables in our study. One intention in this paper was to highlight the surprising, and counter-intuitive pattern showing a positive relationship between elevation and either soil carbon or AGB. As we note in the abstract and introduction, this relationship reflects the presence of EM tree species in Central American montane forests.**

**Therefore, to highlight the importance of soil variables within the context of our elevational gradient and AGB results, we have revised the statement previously made in line 400. The updated version, now found in lines 348–351, reads:** "*It is important to note that temperature, and consequently elevation, is highly correlated with resin P concentration (Fig. 4). This strong correlation complicates the disentangling of the specific contributions of temperature and resin P to the observed patterns—an issue that will be discussed further in a later section.*"

**We additionally added a figure in the main text (Fig. 4) that illustrates the relationship between elevation and: a) resin-extractable phosphorus, b) inorganic nitrogen, and c) the soil total N:P ratio. We also expanded the discussion on the influence of soil fertility, particularly resin P, on AGB (lines 440–449):** "*Evaluating the effects of nutrient availability on AGB along elevational gradients is challenging as direct fertility effects may be masked by the strong correlations between elevation, and therefore temperature, and soil N and P. Neotropical forests are said to transition from being phosphorus-limited in lowland areas (Wright, 2019; Vitousek, 1984; Condit et al., 2013) to nitrogen-limited in montane regions (Wright, 2019; Vitousek, 1984; Tanner et al., 1998) influencing plant functioning and therefore forest productivity. In this study, we found that resin P was highly correlated with elevation, consistent with another elevational gradient in Peru (Nottingham et al., 2015a) and indicating P is not a limiting factor in this montane system. Although there is more P available in montane systems, this does not always result in higher productivity. For instance, in an elevational gradient in Ecuador, forest productivity was influenced not by P or N alone, but by a combination of soil fertility factors, which can outweigh the effects of elevation. (Homeier and Leuschner, 2021).*"

During the sampling and the whole paper, roots are totally neglected. I understand that root sampling has several methodological challenges such as the difficulty to identify the individual/species and root production, as well as the labor-intensive sample processing. But if we don't sample them, we might miss a big part of the picture. Are roots directly correlating with what we are seeing aboveground? Or maybe since they have more nutrients they don't need to invest as much in roots and therefore larger AG lower BG? I am not asking you to repeat your sampling, I still believe it is valuable, but acknowledging that fact somewhere and providing further discussion, citations and further development is necessary.

**Response: Thanks for highlighting this omission from our manuscript. Indeed, roots can provide contribute substantially to carbon storage. We unfortunately lack information from our sites on either fine root dynamics or coarse root biomass. To account for the potential contribution of roots and litter to our total carbon estimates, we have added some sentences about this in the discussion (lines 427–437):** "*Carbon in forests is also stored in pools not included in this study (root biomass and litter), which could contribute significantly to the total carbon stock. Roots biomass estimates are mostly from samples from tropical lowland forests, understating montane ecosystems (Iversen et al., 2021). Recent findings from old-growth montane forests in Africa, where AGC is very high (Cuni-Sanchez et al., 2021), show that tree roots can store up to 59.3 Mg C ha⁻¹ (Yaffar et al., 2023). Values of AGC of Cuni-Sanchez et al (2021) are similar to the ones we found in this study; assuming parity with African montane forests, inclusion of root biomass estimates in our system would increase the*

*total C stock by nearly 9%. Additionally, from a study in Quercus forests at 2900 m elevation in Costa Rica, fine-roots store approximately 5.5 Mg C ha⁻¹ (Hertel et al., 2003), a value three times higher than Andean forest with same elevation (1.79 Mg C ha⁻¹, (Moser et al., 2011). Finally, in montane forests, litter production—including leaves, reproductive organs, twigs, and other components—can range from 2 to 5 Mg C ha⁻¹ y⁻¹ (Moser et al., 2011). This suggests that total C stocks in our plots could increase by up to approximately 1% per year when accounting for litter production.*"

Aligning with my previous point, AM and ECM associations have been done through a database that is generalized among species and there is no mycorrhizal sampling performed in this study. Assuming that we can fully trust that assignation, are we expecting that the colonization % of among species and trees would vary significantly? I don't think this is acknowledged or discussed in the paper. Also, Soudzilovskaia's table for assignation sometimes provides potential open assignations, not fully committed to AM or ECM or even non-mycorrhizal. How did you deal with that uncertainty or other mycorrhizal categories? It is not disclosed in the methodology.

**Response: We now provide more information on the classification of mycorrhizal type in the methods. EM status has been confirmed for the most abundant taxa by Corrales et al. (2016, 2018). We note, however, that the while AM taxa predominate at our site, there are some species that are either non-mycorrhizal (e.g., *Roupala*) or form ericoid mycorrhizal associations (e.g., *Vaccinium*). We classified trees in EM and non-EM groups to include trees that we don't know the mycorrhizae status of, are AM, or do not form an association (lines 107–111): "*Trees were classified as EM (ectomycorrhizal) based on literature sources and root studies made at the field site (Corrales et al., 2018; Soudzilovskaia et al., 2020; Averill et al., 2019; Corrales et al., 2016). Using the same references, trees that have been classified as forming associations with AM, ericoid mycorrhizas (e.g., Vaccinium), or as non-mycorrhizal trees (e.g., Roupala), as well as species for which the mycorrhizal type is unclear, were classified as non-EM taxa.*"**

**Corrales, A., Arnold, A.E., Ferrer, A.H., Turner, B.L., Dalling, J.W. (2016) Variation in ectomycorrhizal fungal communities associated with *Oreomunnea mexicana* (Juglandaceae) in tropical montane forests. Mycorrhiza, doi: 10.1007/s00572-015-0641-8**

**Corrales, A., Henkel, T.W. and Smith, M.E. (2018). Ectomycorrhizal associations in the tropics–biogeography, diversity patterns and ecosystem roles. New Phytologist, 220(4), pp.1076-1091.**

In your explanation, you suggest that the association of species with AM or ECM has a role in wood density (i.e., section 2.5.2). Using an ANOVA to determine that is too simplistic. Is there a phylogenetic bias behind that? For example, gymnosperms are normally associated with AM, and they also normally have lower-density wood. Are you seeing the effect of mycorrhiza or only phylogeny?

**Response: We have very few EM taxa in our plots – Oreomunnea, Quercus, Alfaroa. These are all in a single clade (Fagales). In fact, all the Fagales in these forests are EM. No other**

**taxa that contribute meaningfully to ecosystem carbon form EM associations. So, we do have a phylogenetic effect and the finding that we have higher wood density in ECM species could be a consequence of their mycorrhizal association or of their phylogenetic placement. Incidentally, we have only one coniferous species in our study, *Podocarpus oleifolius*, which only occurs in one plot and forms AM associations. Mean Wood density for Podocarpus +/- 1 SD (0.52 +/- 0.09 ) is similar to the community average for the Fortuna sites (0.57 +/- 0.21)**

I believe it is interesting to see that Lidar underestimates AGB, in Figure 3. Still, this section of the paper seems a bit disconnected from the rest in a way that, for example, it is not mentioned in the abstract and Lidar is not presented in the introduction. Further work should be done to better incorporate this section in the context of the paper.

**Respond: To better incorporate the published LiDAR study and highlight the LiDAR results we included it in the abstract (lines 20–21: "*We found previous LiDAR-derived estimates for our site substantially underestimated the AGC present in the plots which were between two and six times lower than our field-based estimates.*").**

**Specifically, we now add a paragraph on LiDAR to the introduction as follows (lines 86–92): "*In recent decades LiDAR technology has been used broadly to map aboveground carbon stocks at large scale (Saatchi et al., 2011; Zolkos et al., 2013; Carreiras et al., 2017). However, in complex terrains, such as montane tropical forests, LiDAR using low point densities can result in large AGB uncertainty (Leitold et al., 2014), while models of AGB need to specifically represent montane forest structure (González-Jaramillo et al., 2018). We highlight this need by comparing the ground-derived estimates of AGB for our plots with those derived from a combination of airborne Light Detection and Ranging (LiDAR) and more limited ground truthing datasets of Panama that did not account for structural and compositional variation between lowland and montane forests (Asner et al., 2013).*"**

**As well as in the discussion (lines 410–412): "*Using LiDAR sensor in combination with field measurements in montane forests in Ecuador (González-Jaramillo et al., 2018) highlights the importance of terrain features (e.g., ridges, depressions, and protection) in estimates of AGB. For example, ravines, a landscape attribute that is usually ignored at large scales, can have high AGB.*"**

Minor comments:

Line 32: Please, specify what factors.

**Response: We have added these factors (line 33): temperature, nutrient availability and tree mycorrhizal status.**

Line 38: Include a "that" as such: "a meta-analysis found that at lower elevations…"

**Response: Done**

Line 39: It would be nice to specify the reasons why of this low productivity. Do Quesada et al., 2012 provide that? Is it the wood density reasoning that you provide immediately later?

**Response: The sentence has been changed (line 40–41): "*However, low values of AGB have also been found in the Amazon below 500 m a.s.l due to low fertility (Quesada et al., 2012).*"**

Section 2.1. It would be desirable to include a figure that would illustrate such plots, even if it is displayed as SM.

**Response: A figure with a map including the location of the plots has been included in the supplementary materials (Figure S1).**

Line 117-118: I assume the selection of the 13 sampling locations has been randomly selected within the grid because if you divide 1 ha in a grid of 20 x 20 m and sample all of the subplots you do not get 13 samples. Please, specify a bit further about the process.

**Response: To make it clear the sentence has been changed, lines (128–129): "*Soil samples were collected in a regular grid within each 1-ha plot (center of every other 20 x 20 m subplot) resulting in thirteen sampling locations for each plot*".**

Line 121-122: How has this importance been assessed? Did you get this information from the bibliography? If that is the case what papers are those? Alternatively, specify if you had performed any sort of statistical test to determine such variables.

**Response: Citations have been added to support this point.**

Line 146: Do you mean to infer or to generalize instead of "to compare"? I do not fully comprehend what has been done from line 146 to line 151 and why. I guess you want to use the airborne data and the measured biomass to assess how well the two values match and thus make a regional upscaling by using the airborne data?

**Response: Yes, we want to compare the aboveground carbon stock estimated using our ground measurements with values extracted for the same georeferenced locations using LiDAR data. This point is clearer now we have included the topic of LiDAR in the introduction as mentioned in the previous comment.**

Section 2.4.2: If the transects happened every 10m (line 157), it is possible that the same wood debris fell in several transects. For instance, a fallen tree individual that is 30m tall, could easily cross at least 2 transects. What is the protocol for repetitions?

**Response: We understand that a 10 m radius could lead to large fallen trees being recorded multiple times. However, across all plots and 2037 samples, this occurred only 23 times (1.1%). These duplicate records were observed in five plots: Mirador (5), HondaA (7), ChorroA (2), Palos Seco (5), and Hornito (4). We added this sentence in the text to make**

**this clarification (line 301): "*For DCWD, we found that fallen trees were recorded twice on only 23 occasions.*"**

Section 2.5.1 and 2.5.2: I assume the analysis described here has been done in R. Please name the used packages with the respective citation.

**Response: Yes, the R packages have been included for these sections.**

Figure 1: Why some of the vectors are greyer than others? It is not disclosed in the caption.

**Response: The following text has been added to the caption of figure 1: "*Black arrows represent significant effects (p<0.05) and gray arrows non-significant effects*"**

Figure 2a: I suggest including the initials of each plot next to the dots in Figure 2a. This way it would be easier to associate the figure a and b and to translate the % C storage to the total of each fraction.

**Response: We have incorporated the plot initials into this figure.**

Line 285-286: This is interesting. Based on my experience, it is common to sample until 30 cm deep, since it is assumed that those are the most nutrient-dense horizons. Could this finding be a reason to encourage the sampling deeper than that and get until 100cm?

**Response: For these forests with deep carbon-rich soils it is important to sample deeper in the soil profile to fully account for storage. We have added a discussion point for this in line 532–535: "*High carbon content observed at higher elevations suggests that sampling down to 1 m could offer a more comprehensive understanding of nutrient dynamics and carbon stocks in mountainous ecosystems, especially where significant carbon storage occurs below 50 cm.*"**

Figure 7: In addition to the studies citation, I would like to see the locations where they were carried out.

**Response: We have added the site locations to the legend of Figure 7.**

**REFEREE 2 COMMENTS**

Soil and biomass carbon storage is much higher in Central American than Andean montane forests

by Cecilia M. Prada et al

This is well-written manuscript and takes into account most of the available work on carbon storage in tropical montane forests. The questions addressed are interesting and the high C stocks of the studied montane forests are relevant for their conservation.

The comparison with carbon stocks estimated with LIDAR is not well connected to the rest of the study, shoud be better introduced in the introduction. The result, that remote sensing underestimated AGB the importance of ground-truthing with inventories in permanent forest plots should be included in the abstract.

**Response: We have revised the manuscript to better incorporate the published LiDAR study in the abstract, introduction and methods (see response to reviewer 1).**

Table 2 (it contains the same information as presented in Fig. 2) and Fig. 6 (very similar to Fig.1, only needed to explain the variables included in the SEM) both should be moved to the supplement.

**Response: We agree that there is overlap in the information provided in Table 2 and Figure 2. We have therefore moved Table 2 (now Table S1) and Figure 6 (now Fig. S4) to the supplement. However, we prefer to retain Fig 1 and 2 in the manuscript as they convey the major results of the study**

In summary, the manuscript is based on extensive and valid data, but some technical details have to be improved.

Please see some detailed comments below:
28/29: not only coarse woody debris but also the litter layer

**Response: We have changed the sentence (line 28–29) to "*This carbon is sequestered in several pools, consisting of live biomass, necromass (i.e. coarse woody debris or litter), and soil.*"**

35/36: additional reference for soil fertility effects on AGB: Homeier & Leuschner 2021

**Response: This reference has been added to the sentence.**

70-73: additional reference for decreasing nitrogen availability with elevation and effect on wood density: Homeier & Leuschner 2021

**Response: This reference has been added to the sentence.**

97/98: Were tree ferns included?

**Response: Tree ferns were included. This sentence has been edited as (lines 104–105): "*In this study to facilitate comparisons with other published datasets only trees, palms and tree ferns ≥ 10 cm DBH were included in the analyses.*"**

117/118: The litter layer was removed and not part of the soil analyses, do you have any idea of the depth of the litter layer? Fine wood and litter could probably add essential amounts of C to your budgets (e.g. Phillips et al. 2019a).

**Response: Not all plots had a distinct litter layer. When present, we recorded litter depth but did not quantify the carbon content of this pool. We have added a few sentences in the discussion to address the litter layer and other carbon pools that were not included in this study including roots and litter (see our response to Reviewer 1).**

136: In FigS1 it seems that in 3 plots only heights of trees >50cm dbh were available. How does this affect your height estimates?

**Response: Correct - we lack height measurements for smaller trees specifically for these plots. These are all plots at similar elevation at the Fortuna site. To determine how this impacts our height and AGB measurements we used a height-DBH relationship from three sources: 1) plots samples with DBH greater than 50 cm, 2) nearby plots with similar soil characteristics, and 3) pooled height-DBH data from all plots. We applied the Akaike Information Criterion (AIC) to select the best height-DBH model for estimating AGB in each plot. For the Samudio and Alto Frio plots, the best model was the one based on nearby plots with similar soil characteristics (HondaA and Copete respectively). For the Hornito plot, the best model was the one using a pooled height-DBH approach. This way we corrected the height and AGB for Samudio, Alto Frio and Hornito plots. For the Samudio plot, we overestimated AGB by approximately 12% compared to our original data, due to the potential overestimation of tree heights for those with DBH less than 50 cm. On the other hand, for the Alto Frio and Hornito plots, we underestimated AGB by about 10% for both plots because the heights of larger trees were potentially underestimated. We have corrected these values in all data and main text.**

[Figure]

[Figure]

**Figure. Height-DBH models used for estimating AGB for three plots we don't have height samples data from small DBH. a-c) The original data we have used to estimate AGB. d-f) Corrected height used after using the best model for each plot.**

We additionally have added this text in Figure S2 caption: "*Note: Because we do not have height measurements for small trees in the Samudio, Alto Frio, and Hornito plots, we used a height-DBH relationship from three sources: 1) plots samples with DBH greater than 50 cm, 2) nearby plots with similar soil characteristics, and 3) pooled height-DBH data from all plots. We applied the Akaike Information Criterion (AIC) to select the best height-DBH model for estimating AGB in each plot. For the Samudio and Alto Frio plots, the best model was the one based on nearby plots with similar soil characteristics (HondaA and Copete respectively). For the Hornito plot, the best model was the one using a pooled height-DBH approach*"

137: delete "(Fig. S1)"

**Response: We have done this.**

139: Wood volume was quantified with the water displacement method.

**Response: We have edited this sentence (line 149–150) as follows: "*Wood volume in the field was quantified with the water displacement method (Chave, 2006).***"

142/143: Was palm AGB also calculated with the Chave (2014) equation? There are some specific allometric equations for palms: Avalos G et al.(2022), Goodman RC et al.(2013).

**Response: We have incorporated the specific equations for palms suggested (line 157). We also have incorporated an equation for tree ferns (line 161) and AGB was corrected for these taxa: Lines 155–162 "*We used Avalos et al. (2022) allometric equation for Palms aboveground carbon Eq. (2):***

$$AGC = exp(-4.11+1.96*ln(D)+0.8*ln(H)) \qquad (2)$$

***In this case a factor of 2 was used to convert carbon stock to AGB, and a factor of 1.4 was applied to account for the bias introduced by log-transformation, as described by Avalos et al. (2022). Finally we used Beets et al. (2012) allometric equation for tree ferns aboveground carbon Eq. (3):***

$$AGC = 0.0027*(D2*H)1.19 \qquad (3)$$

***A factor of 2 was used to convert carbon stock to AGB following Beets et al, (2012).*"**

174/175: Does that mean that standing dead wood was classified into the same 3 categories as DCWD?

**Response: Yes. We have made this clearer in the manuscript. Lines 191–193: "*In this case p was calculated using the DCWD destructive samples. Since the wood densities differed between DCWD decomposition categories (F(2,326) = 87.35, P = <0.001), wood density was calculated as the average of each decomposition category by plot.*"**

184-191: Was the organic layer included in the sampling?

**Response: No, since not all the plots had an organic layer data, we just used the mineral soil.**

204-206: Fig. 6 should be moved to the supplement. Was the climate axis (PC1 from the climate PCA) not used in the SEM?

**Response: PC1 was used in the analysis and is referred to as "climate parameters" in Figure 6. Additionally, Figure S4 (originally Figure 6) has been moved to the supplementary material.**

213/214: I would suggest to use total CWD instead of only DCWD. Regressions of total CWD should be included in Table S2.

**Response: We chose to remove standing CWD because it was not correlated with the other variables of interest and had a high degree of uncertainty. We retained DCWD for the SEM analysis, but we included the total CWD in Table S2**

238: "WD" in figure legend but "WC" in figure.

**Response: This has been corrected.**

286-289: In the first sentence you say there is no correlation to %EM, in the second sentence you mention there is one correlation. And the C proportion in 10-20 cm depth is negatively correlated with %EM in Fig. S5

**Response: The sentence has been corrected (lines 310–313) as: "*The percent of C in each depth relative to the total C% in each plot was not correlated with % EM, except in the 10-20 cm depth (Fig. S7). For this layer, percent of C was negatively correlated with the percent of basal area contributed by EM-trees (% EM; r2 = 0.36, p < 0.05,), however this effect was not significant after controlling for elevation*"**

326: Only soil parameters of the first PCA axes are included, not hte second axis and no climate parameters.

**Response: We revised the sentence to clarify that we included both soil and climate parameters using the first principal component (PC) axis for each dataset. Lines 356–357 now state: "*Soil and edaphic parameters are represented by the first axis of a PCA for each group of variables where resin P and NH4 were not included.*"**

329: Shouldn't values of relative importance (standardized path coefficients) be between 0 and 1?

**Response:  The model was reviewed and corrected, and updates were made to Figure 6 and the corresponding text in lines 443–4351: "*From the structural equation model (SEM) that included soil, climate variables, AGB, DCWD, and soil C we found temperature was the variable that best predicted AGB and soil C, after controlling for soil and other climate parameters (Fig. 6). Temperature was significantly negatively correlated with AGB and soil C. NH4 and % EM showed a significant correlation with DCWD. Downed CWD decreased significantly with high values of NH4 and increased with high values of % EM. Additionally, % EM was significantly positively correlated with resin P. The effect of climate parameters other than temperature and soil parameters other than NH4 and resin P (represented in the PCA axis, Fig. S4), did not have a significant effect on C pools. It is important to note that temperature, and consequently elevation, is highly correlated with resin P concentration (Fig. 4). This strong correlation complicates the disentangling of the specific contributions of temperature and resin P to the observed patterns—an issue that will be discussed further in a later section.*"**

392/393: Is there any explanation for the extremely high deadwood values in the HA plot?

**Response: In this plot there were two big standing dead oak trees with some of the branches in the subsamples. We added this explanation in lines 421–422: "*One plot, HA, exhibited exceptionally high SCWD biomass (313 Mg ha⁻¹), accompanied by large confidence intervals. This variability was attributed to the presence of two large standing dead oak trees, with some branches included in the subsamples.*"**

406/407: increasing wood density with elevation was only found in some studies, but not in your study

**Response: This comment was correct in lines 452–454: "*Decreasing temperature can reduce growth rates and is associated with increased wood density (Muller-Landau, 2004; Chave et al., 2009), which may increase AGB, however it was not the case in our study*"**

408: elevation and.

**Response: Done.**

427-429: What are the wood densities of these taxa?

**Response: These species exhibit medium to high wood density values. We have added these values, along with those for the oak species, in the text. Lines 474-476: "*We propose that non-EM trees (e.g., Magnolia, Cornus, Cedrela, Cinnamomum), which have wood densities ranging from 0.45 to 0.68, may require similar architectural and wood density traits to coexist with oak species, whose wood densities range from 0.68 to 0.81.*"**

477/478: This sentence is speculative I suggest to delete it.

**Response: Done.**

674: Costa Rica.

**Response: We have corrected this reference.**

Figures and Tables

Fig. 1: What is the difference between the variables with black and grey arrows?

**Response: The following text has been added to the caption of figure 1: "*Black arrows represent significant effects (p<0.05) and gray arrows non-significant effects*"**

"WC" should be "WD" in Fig 1b.

**Response: Done.**

Fig. 4: The size of the boxes in the right-hand column should be the same. I suggest to add also SCWD (or total CWD) as additional box to the left column to show 8 parameters in total.

**Response: We have rearranged now figure 5 and added a plot for SCWD.**

Fig. 6 should be moved to the supplement.

**Response: Figure 6 (now Fig. S4) was moved to the supplementary material.**

Fig. 7a: For comparing with data from Ecuador the Homeier & Leuschner (2012) study with > 80 plots would be the better choice than the Moser et al (2008) study with 5 plots.

**Response: We have added the Homeier & Leuschner (2012) reference and added it to the analysis, including the text and figure 7a.**

Table 2 should be moved to the supplement (it contains the same information presented in Fig. 2).

**Response: We have moved Table 2 (now Table S1) to the supplementary material.**

Table S4: WC?

**Response: We have corrected this text.**